# High Fat High Sucrose Diet Modifies Uterine Contractility and Cervical Resistance in Pregnant Rats: The Roles of Sex Hormones, Adipokines and Cytokines

**DOI:** 10.3390/life12060794

**Published:** 2022-05-26

**Authors:** Róbert Gáspár, Judit Hajagos-Tóth, Annamária Schaffer, Anna Kothencz, Lilla Siska-Szabó, Eszter Ducza, Adrienn Csányi, Tamás Tábi, Fruzsina Bagaméry, Éva Szökő, Orsolya Kovács, Tamara Barna, Reza Samavati, Mohsen Mirdamadi, Anita Sztojkov-Ivanov, Kálmán Ferenc Szűcs, Sandor G. Vari

**Affiliations:** 1Department of Pharmacology and Pharmacotherapy, Albert Szent-Györgyi Medical School, University of Szeged, 6720 Szeged, Hungary; hajagostj@gmail.com (J.H.-T.); ann.schaffer94@gmail.com (A.S.); kothenczanna@gmail.com (A.K.); slilla.szabo@gmail.com (L.S.-S.); kovorsi3@gmail.com (O.K.); barna.tamara@med.u-szeged.hu (T.B.); samavati.reza@med.u-szeged.hu (R.S.); mirdamadi.seyedmohsen@med.u-szeged.hu (M.M.); szucs.kalman@med.u-szeged.hu (K.F.S.); 2Department of Pharmacodynamics and Biopharmacy, Faculty of Pharmacy, University of Szeged, 6720 Szeged, Hungary; ducza.eszter@szte.hu (E.D.); drcsanyi.adrienn@gmail.com (A.C.); sztojkov.ivanov.anita@szte.hu (A.S.-I.); 3Department of Pharmacodynamics, Semmelweis University, 1089 Budapest, Hungary; tabi.tamas@pharma.semmelweis-univ.hu (T.T.); bagamery.fruzsina@pharma.semmelweis-univ.hu (F.B.); szoko.eva@pharma.semmelweis-univ.hu (É.S.); 4International Research and Innovation in Medicine Program, Cedars–Sinai Medical Center, Los Angeles, CA 90048, USA

**Keywords:** obesity, pregnancy, rat, adipokines, cytokines, sex hormones, uterine contractility, cervical resistance

## Abstract

Background: In obesity, the adipose tissue becomes a very significant endocrine organ producing different factors called adipokines, such as leptin, adiponectin and kisspeptin; however, no data are available about their actions on uterine contraction in obese pregnant rats. Our aim was to study the impact of obesity on pregnant uterine contraction in a rat model. Methods: Obesity was induced by the consumption of a high fat high sucrose diet (HFHSD) for 9 weeks, including pregnancy. Glucose tolerance, sex hormone, cytokine and adipokine levels were measured. Uterine contractions and cervical resistance, as well as their responses to adipokines, were tested along with the expressions of their uterine receptors. Results: HFHSD increased body weight, and altered glucose tolerance and fat composition. The uterine leptin and kisspeptin pathway affect increased. The levels of proinflammatory cytokines were reduced, while the plasma level of progesterone was increased, resulting in weaker uterine contractions, and improving the uterine relaxing effects of adipokines. HFHSD reduced cervical resistance, but the core effect of adipokines is difficult to determine. Conclusions: Obesity in pregnant rats reduces uterine contractility and cytokine-induced inflammatory processes, and therefore obese pregnant rat methods are partially applicable for modelling human processes.

## 1. Introduction

There is increasing prevalence and incidence of gestational obesity, which places a significant burden on the health care system and may have a negative epigenetic impact even on children [1]. Gestational obesity increases the incidence of Caesarean section, postpartum hemorrhage, gestational diabetes mellitus, delayed cervical ripening and may lead to congenital abnormalities and diseases in the offspring [2]. The consequences of gestational obesity in uterine contraction and birth weight seem diverse and sometimes controversial. Obesity can reduce the expression of contraction-associated proteins and the contractility of the pregnant uterus in humans and rats [3,4], but it may also increase the risk of preterm birth [5]. Reduced risks for both microsomia and macrosomia were reported in obese pregnancies in humans [6,7], but a reduced birth weight was found in rodents that correlates also with the duration of high calorie feeding [8].

In obesity, the adipose tissue becomes an influential endocrine organ producing different factors called adipokines, such as leptin, adiponectin and kisspeptin [9] and many others. These peptides play an important role in reproduction, embryo development, brain cardiovascular function, glucose homeostasis and lipid metabolism [10]. Furthermore, the enlarged adipocyte undergoes continuous remodeling from hyperplasia to hypertrophy, therefore the synthesis of these adipokines is also altered in obesity [11,12]. The ontogeny of kisspeptin, leptin and adiponectin receptors through the gestation period and the effects of adipokines on uterine contractions have been proved in rats [13,14]. However, no data are available about their effects on uterine contraction in obese pregnant rats. Additionally, gestational obesity modifies the levels of maternal sex hormones and cytokines, which may also have a significant impact on uterine contractility [15,16] and may even contribute to cervical insufficiency [17]. Obesity-induced accumulation of adipose tissue is considered as one of the main reasons for maintaining low grade inflammation, responsible for deleterious pathophysiological processes [18]. Altogether, the adipose tissue in obesity and the produced adipokines, along with sex hormones and cytokine levels, are physiological parameters which change during pregnancy, and which may alter pregnant uterine contractions, leading to contractility disorders. However, no study has been carried out to investigate the complexity of these parameters on uterine contractions.

Our aim was to study the impact of obesity on pregnant uterine contractions in obese rats. Obesity was induced by the consumption of a high fat, high sucrose diet, the altered glucose homeostasis was measured by a glucose tolerance test, and the sex hormone, cytokine and adipokine levels (kisspeptin, leptin, adiponectin) were also measured. Uterine contractions and cervical resistance, as well as their responses to kisspeptin, leptin and adiponectin, were also tested along with the expressions of uterine adipokine receptors.

## 2. Materials and Methods

### 2.1. Housing and Handling of the Animals

The animals were treated in accordance with the European Communities Council Directive (2010/63/EU) and the Hungarian Act for the Protection of Animals in Research (Article 32 of Act XXVIII). All experiments involving animal subjects were carried out with the approval of the National Scientific Ethical Committee on Animal Experimentation (registration number: IV./3071/2016.).

Male and female Sprague–Dawley rats were obtained from Animalab Ltd. (Vác, Hungary) and were housed under controlled temperature (20–23 °C), in humidity (40–60%) and light (12 h light/dark regime)-regulated rooms. After mating, female rats became pregnant and delivered pups. The female offspring were weaned from their mothers at 3 weeks of age, divided into 2 groups (*n* = 12 of each group) and started to feed with a high fat high sugar diet (HFHSD) (C1011, Altromin Spezialfutter GmbH & Co. KG, Lage, Germany) or standard diet (SD) (1314, AltrominSpezialfutter GmbH & Co. KG, Lage, Germany) from 4 weeks of age until the day of sacrifice (12 weeks of age, day 22 of pregnancy), with tap water available ad libitum. The weight of the animals and food consumption were measured weekly. The animals were sacrificed under isoflurane anesthesia using a portable small animal anesthesia machine (R550, RWD, Shenzhen, China), the organs were removed, and wet weights were measured on an analytical balance (Kern ABJ-NM, Kern & Sohn GmbH, Balingen-Frommern, Germany). Blood samples were collected by cardiac puncture in a small animal operating room using sterile syringes, needles and collecting tubes. Plasma samples along with adipose tissue samples (visceral, brown, and gonadal) were stored at −80 °C until the analytical experiments.

### 2.2. Mating of the Rats

The 9-week-old SD and HFSD-fed female (200–250 g) and mature SD-fed male (240–260 g) Sprague-Dawley rats were mated in a special mating cage in the early morning hours. The mature female rats were selected by the estrus cycle. The estrus cycle was measured by an Oestrus Cycle Monitor (IM-01, MSB-MET Ltd., Balatonfüred, Hungary) between 3–4 p.m. Rats with vaginal impedance values between 5.0–8.0 kΩ are in the proestrus phase and were chosen for the mating process for the next morning. In the mating cage, a time-controlled metal door separated the rooms for the male and female animals. The separating door was opened in the early morning (5 a.m.) Within 4–5 h after possible mating, intercourse was confirmed by the presence of a copulation plug or vaginal smears. In positive cases, the female rats were separated, and this was regarded as the first day of pregnancy. During pregnancy, the rats continued their diets, as had been determined before pregnancy.

### 2.3. Glucose Tolerance Test

Glucose tolerance test (GTT) was carried out at 9 weeks of age (6 weeks of HFHSD). Glucose levels were measured with a OneTouch^®^ UltraMini^®^ Glucose Meter (Milpitas, CA, USA). Animals were fasted for 16 h before glucose measurement. Fasting glucose levels were measured first at 8 a.m., then each rat was treated intraperitoneally with 2 mg/kg glucose in the form of a 25% solution. Blood glucose levels were determined 15, 30, 45, 60, 90 and 120 min after the injection. The GTT was evaluated by the comparison of glucose level at each sampling time and the area under curve of the plasma glucose concentration-time curve between the SD and HFHSD groups.

### 2.4. RT-PCR Studies

After sacrifice and excision, the uterine tissues from pregnant rats (*n* = 6) (tissue between two implantation sites) were rapidly placed into RNAlater Solution (Sigma-Aldrich, Budapest, Hungary). The tissues were frozen in liquid nitrogen and stored at −75 °C until the extraction of total RNA.

Total cellular RNA was isolated by extraction with guanidinium thiocyanate-acid-phenol-chloroform according to the procedure described previously [19]. After precipitation with isopropanol, the RNA was washed with 75% ethanol and then re-suspended in diethyl pyro-carbonate-treated water. RNA purity was controlled at an optical density of 260/280 nm with BioSpec-nano (Shimadzu, Kyoto, Japan); all samples exhibited an absorbance ratio in the range of 1.6–2.0. RNA quality and integrity were assessed by agarose gel electrophoresis.

Reverse transcription and amplification of the PCR products were performed by using the TaqMan RNA-to-CT-Step One Kit (ThermoFisher Scientific, Budapest, Hungary) and an ABI StepOne Real-Time cycler. Reverse-transcriptase PCR amplifications were performed as follows: at 48 °C for 15 min and at 95 °C for 10 min, followed by 40 cycles at 95 °C for 15 s and at 60 °C for 1 min. The generation of specific PCR products was confirmed by melting curve analysis. The samples of the PCR experiments contained “no-template” control and “absolute” control or RNA samples from non-treated and treated uteri. The following primers were used: assay ID: Rn01433205_m1 for Ob-R (leptin receptor), Rn01483784_m1 for AdipoR1 (adiponectin receptor 1), Rn01463173_m1 for AdipoR2 (adiponectin receptor 2), Rn00576940_m1 for Kiss1R (kisspeptin receptor), and Rn00667869_m1 for β-actin (ThermoFisher Scientific, Budapest, Hungary) as endogenous control. All samples were run in triplicate. The fluorescence intensities of the probes were plotted against the PCR cycle number. The fluorescence changes were normalized using delta delta Cp method. The amplification cycle displaying the first significant increase in the fluorescence signal was defined as the threshold cycle (CT).

### 2.5. Western Blot Analysis

After sacrifice and excision, the uterine tissues from pregnant animals (*n* = 6) (tissue between two implantation sites) were homogenized using a Micro-Dismembrator (Sartorius AG, Goettingen, Germany) and centrifuged at 5000× *g* for 15 min at 4 °C in RIPA Lysis Buffer System (Santa Cruz Biotechnology, Inc., Dallas, TX, USA), which contained phenyl-methyl-sulfonyl fluoride, sodium orthovanadate and protease inhibitor cocktail. Total protein amounts from the supernatant were determined by a spectrophotometer (BioSpec-nano, Shimadzu, Japan).

Twenty-five micrograms of sample protein per well was subjected to electrophoresis on 4–12% NuPAGE Bis-Tris Gel in XCellSureLock Mini-Cell Units (Life Technologies, Budapest, Hungary). Proteins were transferred from gels to nitrocellulose membranes using the iBlot Gel Transfer System (Life Technologies, Budapest, Hungary). Ponceau S (Sigma-Aldrich, Budapest, Hungary) was used to check the standard running and transfer conditions. The blots were incubated overnight on a shaker with Kiss1R (54 kDa), AdipoR1 (49 kDa), AdipoR2 (44 kDa), Ob-R (100/125 kDa) and β-actin (42 kDa) antibodies (Santa Cruz Biotechnology, Inc., Dallas, TX, USA) in blocking buffer. The incubation of the secondary antibody solution was carried out based on the protocol of the WesternBreeze^®^ Chromogenic Immunodetection Kit (Thermo Fisher, Waltham, MA, USA). Images were taken with the EDAS290 imaging system (Kodak Ltd., Budapest, Hungary), and the optical densities of immunoreactive bands were determined with Kodak 1D Images analysis software. β-actin was used for protein normalization for this semi-quantitative method. Optical densities were calculated as arbitrary units after local area background subtraction.

### 2.6. Uterine Contractility Studies

The uteri were removed from the SD and HFHSD-fed 22-day pregnant rats. 5 mm-long muscle rings were sliced from both horns of the uterus and mounted vertically in an organ bath containing 10 mL of de Jongh solution (composition: 137 mM NaCl, 3 mM KCl, 1 mM CaCl_2_, 1 mM MgCl_2_, 12 mM NaHCO_3_, 4 mM NaH_2_PO_4_, 6 mM glucose, pH = 7.4). The temperature of the organ bath was maintained at 37 °C, and carbogen (95% O_2_ + 5% CO_2_) was perfused through the bath. After mounting, the rings were allowed to equilibrate for approximately 60 min before the experiments were started, with a buffer change every 15 min. The initial tension of the preparation was set to about 1.5 g and the tension dropped to about 0.5 g by the end of the equilibration period. The tension of the myometrial rings was measured with a gauge transducer (SG-02; MDE Ltd., Budapest, Hungary) and recorded with a SPEL Advanced ISOSYS Data Acquisition System (MDE Ltd., Budapest, Hungary). In the following step, spontaneous contractions were recorded, then cumulative concentration–response curves were constructed by oxytocin (10^−12^–10^−8^ M) (Gedeon Richter Plc., Budapest, Hungary) or prostaglandin F_2α_ (PGF_2α_, 10^−10^–10^−5^ M) (Sigma-Aldrich Ltd., Budapest, Hungary) in the presence of kisspeptin (KISS1 94-121 fragment, 10^−7^ M) (Sigma-Aldrich Ltd., Budapest, Hungary), leptin (10^−8^ M) (PeproTech EC, Ltd., London, United Kingdom) or adiponectin (10^−9^ M) (Sigma-Aldrich Ltd., Budapest, Hungary). Following the addition of each concentration of contracting agent, recording was performed for 300 s. Concentration–response curves were fitted, areas under curves (AUC) were evaluated and Emax and EC50 values were calculated.

### 2.7. Measurement of Cervical Resistance

Cervical tissues were removed from the 22-day pregnant rats. The two cervical rings were separated and mounted, with their longitudinal axis vertical, on hooks in an organ bath containing 10 mL of de Jongh buffer. The lower sides of the cervices were fixed to the bottom of the tissue holders in the organ chambers, whereas the upper parts were hooked to gauge transducers (SG-02; MDE Ltd., Budapest, Hungary). After mounting, the rings were allowed to equilibrate for approximately 1 h before the experiments (incubation period) were carried out, with a buffer change every 15 min. The initial tension was set to approximately 2.00 g.

After incubation, cervical resistance was investigated by gradually increasing the tension in the tissues as described earlier [20]. The precise initial tension and the relaxation of the cervices were followed with an online computer, using the S.P.E.L. Advanced Isosys Data Acquisition System (MDE Ltd., Budapest, Hungary). When drug effects were investigated, kisspeptin (10^−7^ M), leptin (10^−8^ M) and adiponectin (10^−9^ M) were added to the organ bath and the cervices were incubated for 20 min before stretching.

### 2.8. Measurements of Sex Hormones, Adipokine and Cytokine Levels in Plasma and Fat Tissues

Plasma levels of kisspeptin, leptin, adiponectin and 17β-estradiol, progesterone were assayed by ELISA kits from R&D Systems (Minneapolis, MN, USA), Biomatik (Kitchener, ON, Canada) and Cayman Chemical (Ann Arbor, MI, USA), respectively, according to the recommendation of the manufacturer.

For the determination of inflammatory cytokines, TNFα, IL-1β and IL-6 in adipose tissue visceral, subcutaneous, and brown fat samples were homogenized in 10 volumes of cold 100 mM phosphate buffer pH 7.4 by a blade homogenizer. The crude extract was then centrifuged at 20,000× *g*, 4 °C for 10 min and the clear aqueous phase was used for ELISA measurements using kits from R&D Systems according to the recommendation of the manufacturer. The protein content of tissue extracts was measured by the Bradford method and used to correct the ELISA results [21]. All ELISA measurements were performed on a Multiskan EX microplate spectrophotometer (ThermoFisher Scientific, Waltham, MA, USA).

Plasma cytokine levels were determined by digital immunoassays with Single Molecule Arrays (Simoa) using Planar Array Rat Cytokine Panel 1 (Quanterix Corp., Billerica, MA, USA) for seven rat cytokines: interferon gamma (IFNγ), IL-1β, IL-2, IL-6, IL-10, keratinocytes-derived chemokine (KC), and Tumor Necrosis Factor alpha (TNFα). The Planar Panel included sample diluent buffer and assays were performed following the manufacturer protocols. The Panel was analyzed by the SP-X Imaging and Analysis System (Quanterix Corp.) All samples were run in duplicate.

## 3. Results

### 3.1. Effect of HFHSD on Body and Organ Weights, Glucose Tolerance and Fat Distribution in Pregnant Rats

HFHSD significantly increased the body weight of female rats after 3 weeks (6 weeks of age) as compared with the SD group. The higher body weight remained after 6 weeks of diet (9 weeks of age) in HFHSD. From 10 weeks of age, the rats became pregnant, and at the end of pregnancy (22 days of gestation, 12 weeks of age), the body weight difference disappeared between the SD and HFHSD groups (Figure 1).

The food consumption was not different between the SD and HFHSD groups considering the 3-week averages (Figure 2).

On pregnancy day 22, HFHSD significantly reduced uterine, placental, and fetal weights, while increasing the fetal/placental ratio. The fetal number and the skeletal muscle mass remained unchanged (Table 1).

The 9 weeks of HFHSD increased the initial blood sugar level of the rats and increased the blood sugar levels after 60 and 90 min in the glucose tolerance test (GTT). (Figure 3a). However, the area under curve (AUC) analysis of the blood sugar plasma level did not show significant elevation in the HFHSD group as compared to the SD group (Figure 3c). At the end of pregnancy (pregnancy day 20), the repeated GTT showed that HFHSD did not affect the initial glucose level but worsened glucose tolerance in the first 45 min after glucose administration (Figure 3b). The AUC analysis revealed an increased plasma glucose level during GTT in the HFHSD group as compared to the SD group (Figure 3d).

The 9 weeks of HFHSD modified the fat distribution in pregnant rats. The mass of pink fat was dominant in both SD and HFHSD-fed rats on the last day of pregnancy. HFHSD significantly enhanced the weights of gonadal, visceral, pink, and brown adipose tissues along with the total fat mass. Visceral fat showed the greatest gain in weight with HFHSD. Furthermore, HFHSD reduced the ratio of pink adipose tissue originally increased due to the lactation and enhanced the proportions of visceral and gonadal fats (Figure 4).

### 3.2. Effect of HFHSD on the Expressions of Kisspeptin, Leptin and Adiponectin RECEPTORS in Pregnant Rats

The 9 weeks of HFHSD modified the expressions of kisspeptin and leptin receptors in the 22-day pregnant uteri. The mRNA and protein expressions of kisspeptin receptors were significantly increased in the HFHSD group as compared with the SD group (Figure 5). Similar enhancements were detected in the case of leptin receptor (Figure 6). However, HFHSD did not modify the uterine expressions of either adiponectin 1 or adiponectin 2 receptors (Figure 7). The gel pictures of the Western blot measurements are available in the Appendix A.

### 3.3. Effect of HFHSD on Contractility of 22-Day Pregnant Rat Uterus In Vitro

Spontaneous contractility was significantly reduced in HFHSD uteri as compared with the SD group. The area under curve comparison of the 30-min recorded contractions showed a 16% decrease in the HFHSD group (Figure 8).

The diet did not influence oxytocin-induced contractions but modified the response to kisspeptin and leptin. In the SD group, the presence of kisspeptin reduced oxytocin-induced contractions, while HFHSD ceased this effect (Figure 9a). In contrast, the presence of leptin enhanced oxytocin-elicited contractions, but this effect was also suspended by HFHSD (Figure 9b). However, adiponectin did not modify oxytocin contractions in either SD or HFHSD uterine samples (Figure 9c).

Similarly, the diet did not influence PGF_2a_-induced contractions but modified the response to kisspeptin and adiponectin. In the SD group, the presence of kisspeptin reduced PGF_2α_-induced contractions, while HFHSD resulted in a further decrease in this action (Figure 10a). The presence of adiponectin did not modify PGF_2α_-elicited contractions but reduced the contractions in HFHSD uteri (Figure 10c). However, leptin did not modify PGF_2α_ contractions in either SD or HFHSD uterine samples (Figure 10b).

### 3.4. Effect of HFHSD on Cervical Resistance of 22-Day Pregnant Rats In Vitro

HFHSD reduced cervical resistance of 22-day pregnant rats as compared with the SD group. The presence of kisspeptin or adiponectin induced a further decrease, while leptin itself did not modify the cervical resistance of SD samples. However, HFHSD ceased the cervical resistance decreasing the effects of kisspeptin and adiponectin, but triggered the reducing effect of leptin (Figure 11).

### 3.5. Effect of HFHSD on Sex Hormone, Adipokine and Cytokine Levels in 22-Day Pregnant Rats

In plasma, HFHSD increased the level of P4 as compared with the SD group, while the level of E2 remained unchanged (Figure 12a). Among adipokines, only the leptin level was increased by HFHSD (Figure 12b). Among cytokines, the plasma levels of IL-10, KC and TNFα were reduced by HFHSD, while the Il-2 level remained unchanged (Figure 12b). Additional cytokines tested (Il-1β, IL-6, IFNγ) were not detectable in the plasmas of SD or HFHSD-fed rats.

In the adipose tissues, the amounts of TNFα, IL-1β and IL-6 were separately measured in visceral, gonadal, and brown fats. HFHSD reduced the amount of TNFα and IL-6 in brown and gonadal fats, while their visceral fat content remained unchanged as compared with the SD group (Figure 13a,c). The amount of IL-1β was decreased in visceral and gonadal fats with no change in brown adipose tissue (Figure 13b).

## 4. Discussion

In our study, we investigated the impact of obesity on pregnant uterine contractility in rats. Obesity was induced by HFHSD, which is an accepted method for obesity models [22]. The 9 weeks (including 3 weeks of gestation) of HFHSD increased body weights and reduced glucose tolerance as compared with the control group. These modifications confirm the efficacy of HFHSD regarding the development of obesity and prediabetic condition [23]. Additionally, HFHSD reduced uterine, placental, and fetal weights with an increase in the fetal/placental ratio. These kinds of alterations were described earlier [8] and further strengthen that the HFHSD is successfully inducing obesity. Furthermore, the higher fetal/placental ratio along with the reduced uterine size predict a weaker fetal supply and maintenance in pregnant obese rats. These modifications explain why body weights were not significantly different between the SD and the HFHSD groups on the last day of pregnancy: the increase in body weights of the HFHSD group was probably hidden by the reduced weights of the fetuses, placentas, and uteri. Since the average number of pups in a pregnant rat uterus is between 6–10, the measured reduction in their weights may significantly influence the total body weight of a pregnant animal.

HFHSD also modified the total fat weight and the distribution of different kinds of adipose tissue. Although the ratio of brown adipose tissue remained the same, visceral, and pink adipose tissues showed a significant, while gonadal adipose tissue had a moderate, increase. Additionally, the ratio of pink adipose tissue in the mammary gland derives from the trans differentiation of subcutaneous white adipocytes was reduced as compared to visceral or gonadal fats. Pink fat is known to be crucial for the maintenance of lactation [24], thus obesity not only reduces the size and the intrauterine fostering of fetuses due to reduced placental weight, but also creates a worse condition for postnatal nurture in rats. The increase in the mass of brown adipose tissue also reflects the increased heat production [25] to compensate for the HFHSD-induced changes in metabolism. These alterations also confirm the efficacy of HFHSD in the stimulation of food-induced obesity during pregnancy and the appropriateness of our model.

From among the several known adipokines, we focused on three peptides and their role in uterine contractions. Leptin is one of the main adipokines produced by adipose tissue, and its high level participates in low-grade inflammatory processes during obesity [26]. The uterine action of leptin has been clarified in human and rat studies [27,28]. Currently, the ontogeny of leptin and adiponectin in uterine action has been revealed during pregnancy in rats, proving their relaxing effects, although neither of them had an effect on uterine contractions at term [14]. Adiponectin is considered as an anti-inflammatory adipokine balancing the obesity-induced oxidative processes [29]. Kisspeptin is better known as a neuropeptide participating in the control of reproductive processes [30], but in obesity this peptide behaves as an adipokine rather than a neuropeptide [31]. The relaxing effect of kisspeptin on uterine contractions has been identified in pregnant rats along with the ontogeny of the kisspeptin receptor [13].

In our recent study, we found that HFHSD increased the expressions of kisspeptin and leptin receptors but did not alter adipokine receptors in 22-day pregnant uteri. Among adipokines, only the leptin plasma level was increased. Since both the increase in receptor expression and plasma level can enhance the effect of a given adipokine, these results suggest that the leptin pathway has the strongest, while the adiponectin pathway has the weakest (or neutral), influence on HFHSD-induced modifications in uterine contractions. Our previous studies revealed that both leptin and kisspeptin have a relaxing effect on last-day pregnant uteri [13,14], therefore we anticipated that these alterations should reduce uterine contractions.

Additionally, HFHSD significantly increased the progesterone level of 22-day pregnant rats, which also supports uterine quiescence [32], but contrasting with earlier results, confirming that a high estrogen level, but not progesterone, increases leptin receptor expression during pregnancy [33]. Our results suggest that the expression of leptin is under a multifactorial control and is not determined solely by the sex hormone ratio in rats.

Higher fat mass usually predicts higher levels of inflammatory cytokines [34]; however, we found that, although the plasma level of anti-inflammatory cytokine IL-10 was reduced, the levels of two other proinflammatory cytokines (KC and TNFα) were also decreased. There was also significant reduction in the levels of proinflammatory cytokines (IL-1β, IL-6 and TNFα) in adipose tissues. Overall, the sum alteration of these cytokines rather suggests the predominance of anti-inflammatory courses. The changes in adipose tissue are contradictory, since the ratio of visceral, and pink adipose tissues showed significant increase. Nevertheless, it must show a correlation to the metabolic activities of the visceral adipocytes, more in the reproductive phase (hyperplasia) than in the reduced metabolic state (hypertrophy).

The HFHSD-induced changes in sex hormones, adipokines and cytokines projected the alterations in uterine contractility. Spontaneous contractions were reduced, while basic responses to oxytocin or PGF_2α_ did not change, suggesting that obesity may reduce uterine contractions (due to the high progesterone and leptin and low proinflammatory cytokine levels) but does not influence the stimulability of the last-day pregnant rat uterus. Earlier high-fat, high-cholesterol diet-induced obesity was suspected to induce uterine dystocia in a rat model [4], and we observed the same in our recent findings.

We also investigated the uterine actions of adipokines against oxytocin or PGF_2α_-induced contractions. These contractile agents were chosen because of their roles in physiological pregnant uterine contractions and cervical maturation at term [35].

In the case of oxytocin-induced contractions, HFHSD ceased the relaxing and contracting effects of kisspeptin and leptin, respectively. The contracting effect of leptin in SD uteri was not detected in our previous study, but in that measurement the contractions were elicited by KCl [14]. Earlier it was reported that leptin did not modify spontaneous and oxytocin-induced contractions on either human or rat myometria [28], although another experiment proved its relaxing effect on the human uterus [27]. A study proved that leptin has a utero-relaxant effect on spontaneous and oxytocin-induced contractions via nitric oxide (NO) and cyclic guanosine monophosphate (cGMP) pathways in late pregnant mice [34]. Thus, the oxytocin-induced contraction increasing effect of leptin is unusual and may be a rat-specific phenomenon. Nonetheless, this strange effect was blotted out by obesity. In addition, adiponectin did not elicit any action on either SD or HFHSD uteri during oxytocin stimulation.

In the case of PGF_2α_-induced contractions, the uterine relaxing effect of kisspeptin was stronger in HFHSD uteri. Adiponectin elicited a relaxing effect only in the HFHSD group. Conversely leptin did not show any effect on PGF_2α_-induced uterine contractions on the last day of the gestation period. The effects of these adipokines in SD rats are in harmony with our earlier experiments [13,14]. The HFHSD-induced relaxing effect of adipokines further confirms obesity-induced uterine dystocia.

Cervical resistance and ripening are also crucial for pregnancy maintenance and delivery. The changes in the ratio of proinflammatory and anti-inflammatory cytokines can dysregulate the ripening process [36]. Although obesity in humans is known to inhibit cervical ripening due to reduced sensitivity to prostaglandins [37], no information is available about the action of adipokines on cervical resistance. We found that kisspeptin and adiponectin reduced cervical resistance in SD rats, while leptin had no influence at all. Contrary to human experience, obesity reversed the cervical action, reducing the cervical resistance in rats that were increased by kisspeptin. Adiponectin did not elicit any action on HFHSD cervices, while leptin had a resistance reducing effect. Hence, the entire impact of obesity and adipokines on cervical resistance is difficult to determine due to their complexity. Nevertheless, we can make an educated guess that the positive and negative changes compensate each other in last-day pregnant rats.

On the last day of pregnancy in rats, the HFHSD-induced increase in body weight worsened glucose tolerance and altered fat composition. Among the investigated adipokines, the impact of the leptin and kisspeptin pathway has been increased by plasma level or receptor expression, or both. However, the increase in body weight reduced the plasma and adipose tissue levels of proinflammatory cytokines and increased the plasma levels of uterine relaxing progesterone, decreasing the development of low-grade inflammation. These changes resulted in weaker uterine contractions and, in most cases, improved the uterine relaxing effects of the adipokines. HFHSD reduced cervical resistance, but in the presence of adipokines cervical resistance was modified in both directions, therefore making it difficult to determine the aggregate effects. Weight gain in pregnant rats reduces uterine contractility by adipokines and reduces cytokine-induced inflammatory processes. While the former is in harmony with the human results, the latter is contradicts them. These findings suggest only a partial applicability of obese pregnant rat methods for modelling human courses during pregnancy.

## Figures and Tables

**Figure 1 life-12-00794-f001:**
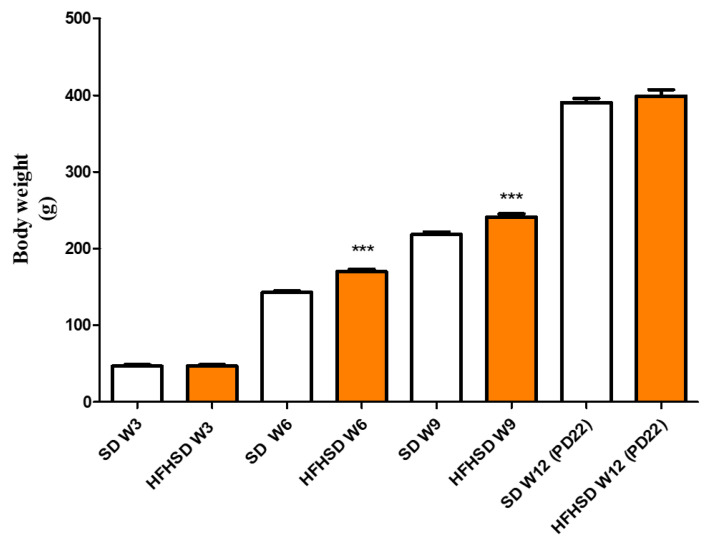
Body weights of standard diet (SD) and high fat high sucrose diet (HFHSD)-fed rats at every 3 weeks (W) of age including the last (day 22) of pregnancy (PD22). The HFHSD diet was started at 4 weeks of age. The rats became pregnant at 10 weeks of age. The statistical significance represents the differences between the SD and HFHSD groups at the same weeks of age. ***: *p* < 0.001.

**Figure 2 life-12-00794-f002:**
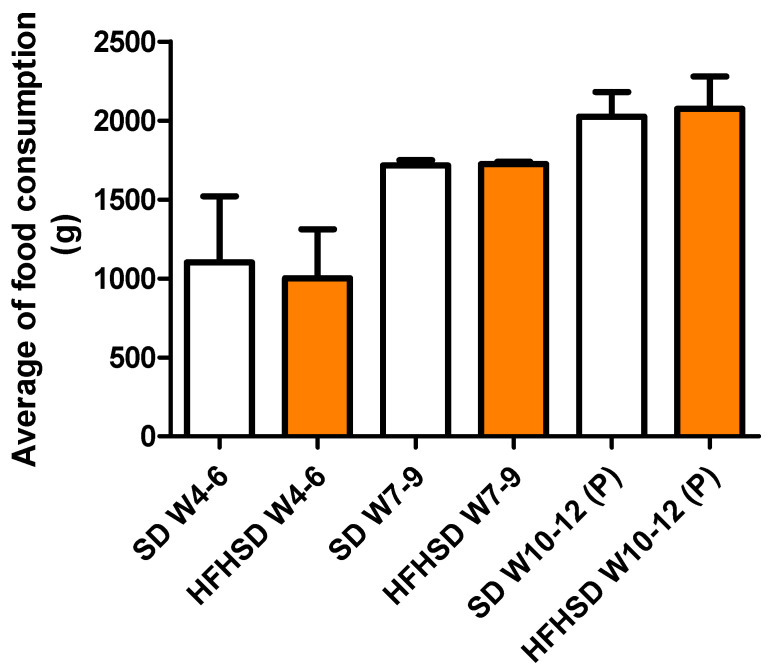
Consumed amount of standard diet (SD) and high fat high sucrose diet (HFHSD) foods expressed as a 3-week average (W) in rats. The last 3 weeks were the pregnant period (P). The HFHSD diet was started at 4 weeks of age. No statistical differences were found between the groups.

**Figure 3 life-12-00794-f003:**
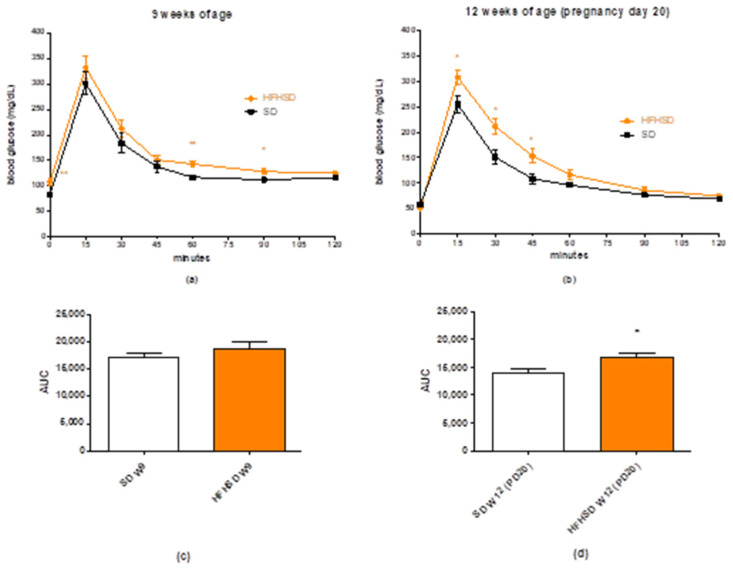
Changes in plasma glucose levels during the glucose tolerance test (GTT) in standard diet (SD) and high fat high sugar diet (HFHSD)-fed rats at 9 and 12 weeks of age (W) (**a**,**b**). The rats become pregnant at 10 weeks of age, and the GTT was carried out on gestational day 20 (PD20). Each time point was compared between the SD and HFHSD groups. The plasma curves were compared by the calculation of area under curve (AUC) (**c**,**d**). HFHSD worsened glucose tolerance after 9 weeks of feeding. The statistical significance represents the differences between the SD and HFHSD groups at the same weeks of age *: *p* < 0.05; **: *p* < 0.01.

**Figure 4 life-12-00794-f004:**
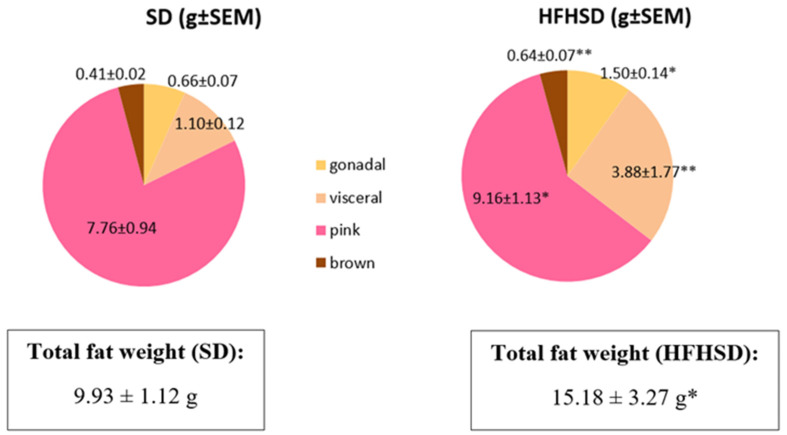
Changes in fat distribution in standard diet (SD) and high fat high sugar diet (HFHSD) groups on the last day of pregnancy (12 weeks of age), after 9 weeks of feeding. Weights of all types of fat tissues and total fat weight were significantly increased. *: *p* < 0.05; **: *p* < 0.01.

**Figure 5 life-12-00794-f005:**
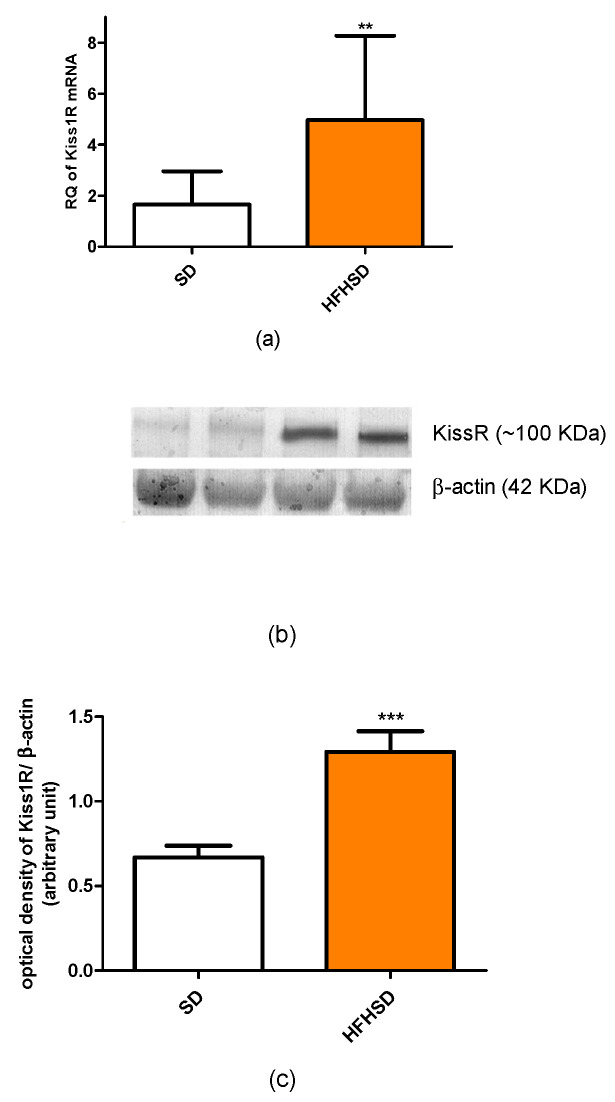
Changes in kisspeptin receptor (Kiss1R) mRNA (**a**) and protein (**b**,**c**) expressions in the uteri of standard diet (SD) and high fat high sugar diet (HFHSD)-fed pregnant rats on the last day of pregnancy (12 weeks of age), after 9 weeks of feeding (*n* = 6). Both mRNA and protein expressions were significantly increased by HFHSD. **: *p* < 0.01; ***: *p* < 0.001.

**Figure 6 life-12-00794-f006:**
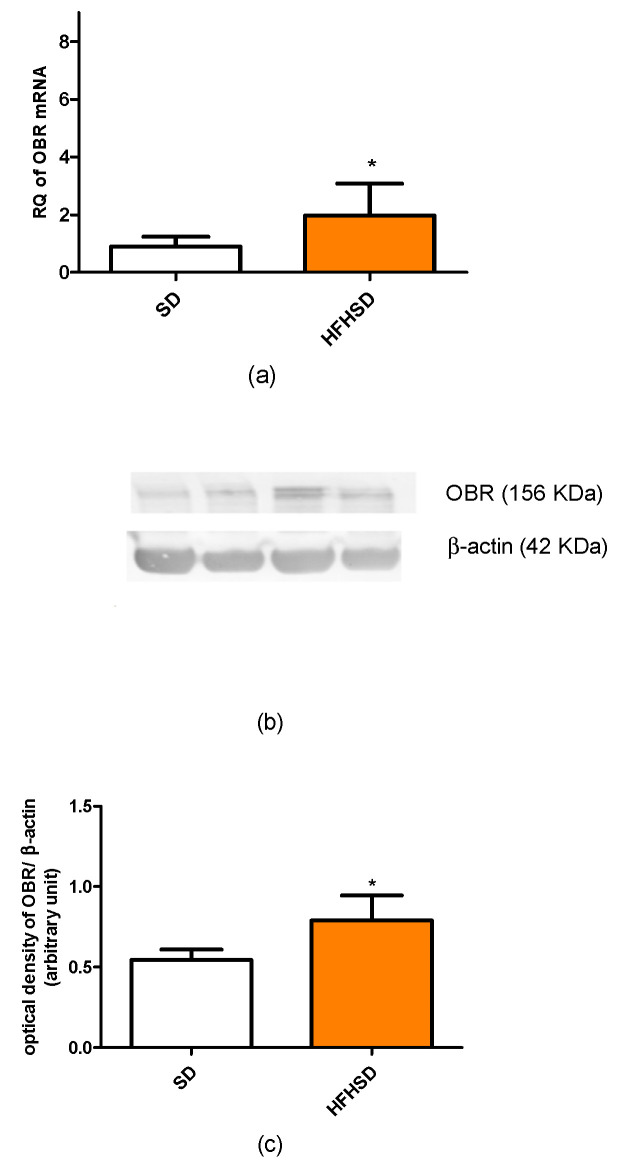
Changes in leptin receptor (OBR) mRNA (**a**) and protein (**b**,**c**) expressions in the uteri of standard diet (SD) and high fat high sugar diet (HFHSD)-fed pregnant rats on the last day of pregnancy (12 weeks of age), after 9 weeks of feeding (*n* = 6). Both mRNA and protein expressions were significantly increased by HFHSD. *: *p* < 0.05.

**Figure 7 life-12-00794-f007:**
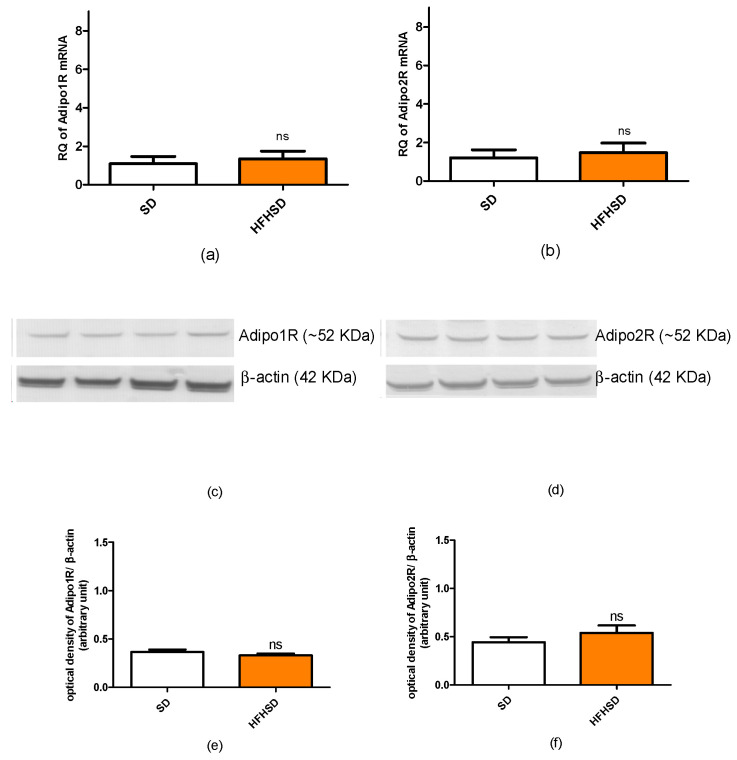
Changes in adiponectin receptors 1 and 2 (Adipo1R and Adipo2R) mRNAs (**a**,**b**) and protein (**c**–**f**) expressions in the uteri of standard diet (SD) and high fat high sugar diet (HFHSD)-fed pregnant rats on the last day of pregnancy (12 weeks of age), after 9 weeks of feeding (*n* = 6). No significant modification was found in any of the receptors. ns: not significant.

**Figure 8 life-12-00794-f008:**
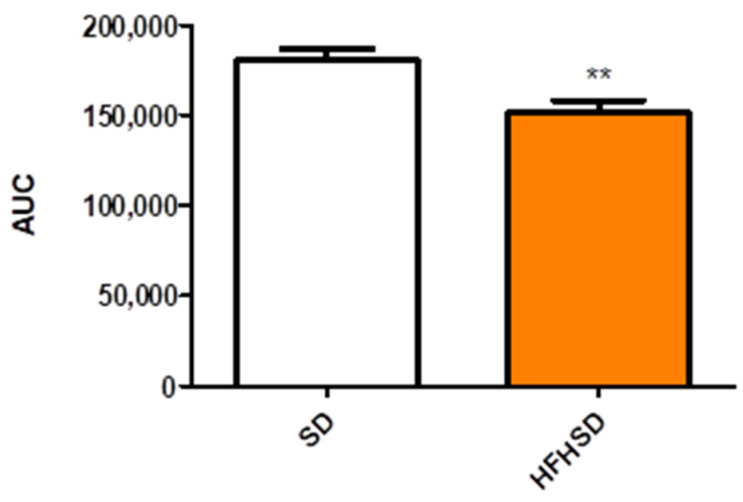
Area under curve (AUC) of spontaneous uterine contractions in standard diet (SD) and high fat high sugar diet (HFHSD)-fed, 22-day pregnant rats in an isolated organ bath system. HFHSD uteri had weaker contractions as compared with SD uteri **: *p* < 0.01.

**Figure 9 life-12-00794-f009:**
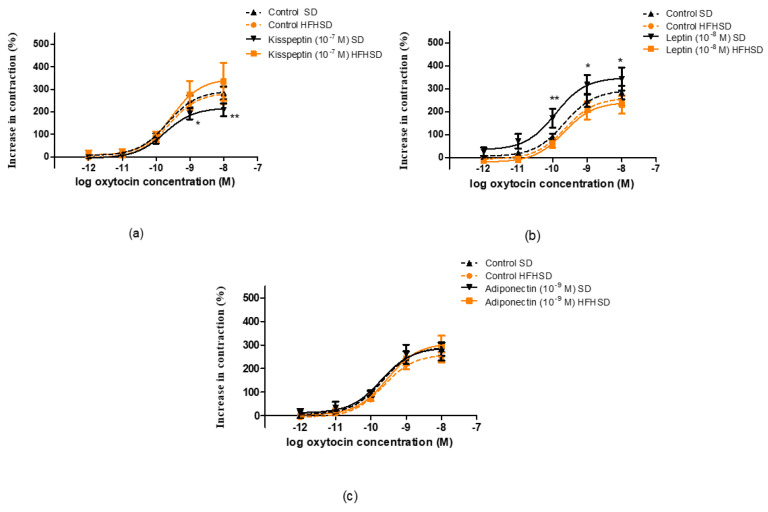
Impact of diet on the effects of kisspeptin, leptin and adiponectin on oxytocin-induced contractions in 22-day pregnant rat uteri. The measurements were carried out in an isolated organ bath system. The high fat high sugar diet (HFHSD) did not modify the contraction response of pregnant uteri to oxytocin as compared with standard diet (SD) samples (dotted lines). Kisspeptin reduced oxytocin contractions in SD rats, which were ceased by HFHSD (**a**), while leptin increased the contractions in SD samples, which were also ceased in HFHSD rats (**b**). Adiponectin did not modify the contractions, and the diet did not change the effect of adiponectin either (**c**) *: *p* < 0.05; **: *p* < 0.01.

**Figure 10 life-12-00794-f010:**
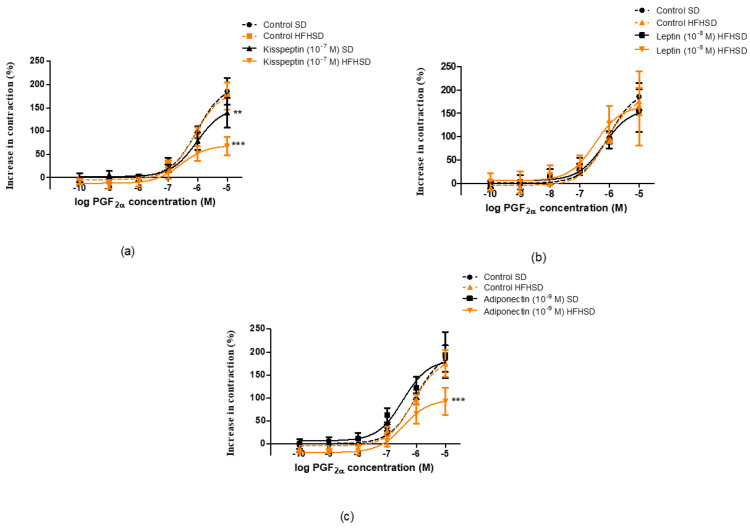
Impact of diet on the effects of kisspeptin, leptin and adiponectin on prostaglandin F_2α_ (PGF_2α_)-induced contractions in 22-day pregnant rat uteri. The measurements were carried out in an isolated organ bath system. The high fat high sugar diet (HFHSD) did not modify the contraction response of pregnant uteri to PGF_2__α_ as compared with standard diet (SD) samples (dotted lines). Kisspeptin reduced PGF_2α_ contractions in SD rats, which was further decreased by HFHSD (**a**). Leptin did not modify the contractions and the diet did not modify the effect of leptin either (**b**). Adiponectin did not modify the contractions in SD samples, but significantly reduced PGF_2α_ contractions in HFHSD rats (**c**) **: *p* < 0.01; ***: *p* < 0.001.

**Figure 11 life-12-00794-f011:**
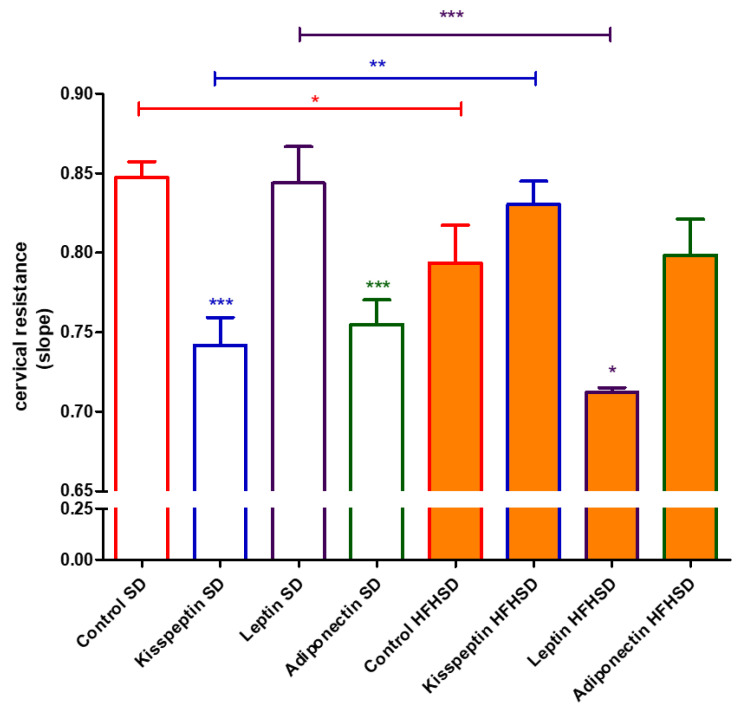
Impact of diet and adipokines on cervical resistance in 22-day pregnant rats. Cervical resistance was measured in an isolated organ bath applying a stretching test. The high fat high sugar diet (HFHSD) itself reduced cervical resistance as compared with the standard diet (SD) in the control group. Kisspeptin (10^−7^ M) and adiponectin (10^−9^ M) reduced cervical resistance in SD samples, while leptin (10^−8^ M) was ineffective. HFHSD ceased the cervical resistance reducing effects of kisspeptin and adiponectin, but leptin further decreased resistance *: *p* < 0.05; **: *p* < 0.01; ***: *p* < 0.001.

**Figure 12 life-12-00794-f012:**
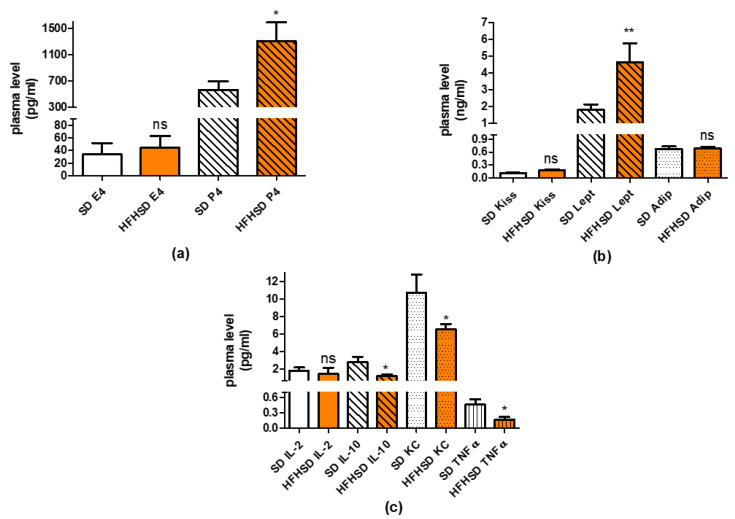
Changes in plasma levels of sex hormones (**a**), adipokines (**b**) and cytokines (**c**) in 22-day pregnant rats induced by high fat high sugar diet (HFHSD, orange columns). The 17β-estradiol (E2, empty columns) level did not change, but the progesterone (P4, cross-striped columns) level was increased as compared to the standard diet (SD, white columns) group. The plasma level of leptin (Lept, cross-striped columns) was also increased, while the levels of kisspeptin (Kiss, empty columns) and adiponectin (Adip, dotted columns) remained unchanged. The levels of interleukin 10 (IL-10, cross-striped columns), keratinocytes-derived chemokine (KC, dotted columns) and tumor necrosis factor alpha (TNFα, vertical-striped columns) were reduced, while the level of interleukin 2 (IL-2, empty columns) was not modified *: *p* < 0.05; **: *p* < 0.01; ns: not significant.

**Figure 13 life-12-00794-f013:**
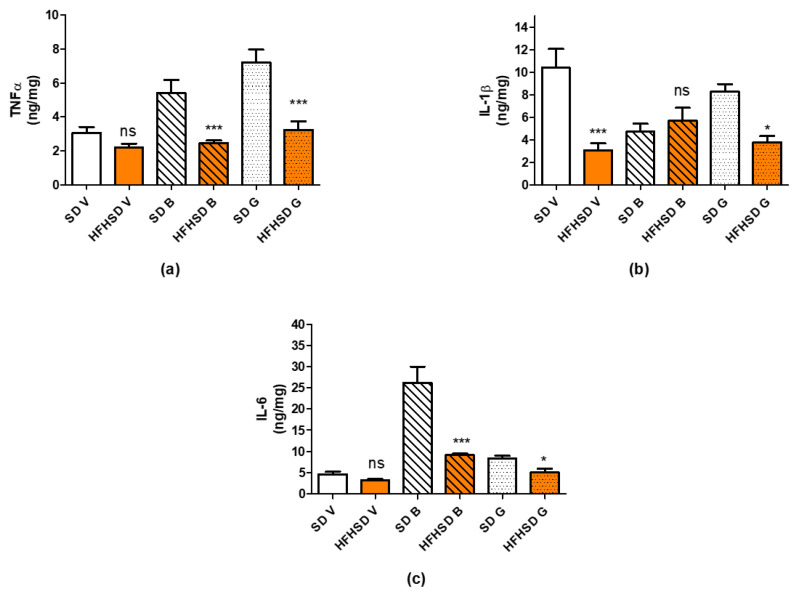
Changes in adipose tissue levels of tumor necrosis factor alpha (TNFα) (**a**), interleukin 1β (IL-1β) (**b**) and interleukin 6 (IL-6) (**c**) in 22-day pregnant rats induced by high fat high sugar diet (HFHSD, orange columns). The amount of TNFα was reduced in brown (B, cross-striped columns) and gonadal (G, dotted columns) adipose tissues, while it did not change in visceral (V, empty columns) fat as compared to the standard diet (SD, white columns) group. The amount of Il-1β was reduced in all fat types except brown. The amount of IL-6 was reduced in brown and gonadal fats *: *p* < 0.05; ***: *p* < 0.001; ns: not significant.

**Table 1 life-12-00794-t001:** Changes in organ weights and fetal numbers induced by high fat high sugar diet in 22-day pregnant rats. SD: standard diet group; HFHSD: high fat high sugar diet group; *: *p* < 0.05, **: *p* < 0.01; ***: *p* < 0.001; n.s.: not significant.

Organ	SD	HFHSD	*p*
uterus (g ± SEM)	4.22 ± 0.13	3.65 ± 0.15	*
placenta (g ± SEM)	0.54 ± 0.01	0.41 ± 0.02	***
fetus (g ± SEM)	5.83 ± 0.10	5.17 ± 0.13	***
fetus/placenta ( ± SEM)	10.93 ± 0.28	12.21 ± 0.27	**
fetal number (n ± SEM)	12.69 ± 0.58	12.84 ± 0.93	n.s.
m. gastrocnemius (g ± SEM)	3.29 ± 0.12	3.51 ± 0.16	n.s.

## Data Availability

Research data are available for request.

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
