# Peer review of "High Fat High Sucrose Diet Modifies Uterine Contractility and Cervical Resistance in Pregnant Rats: The Roles of Sex Hormones, Adipokines and Cytokines"

_life, 2022, doi:10.3390/life12060794_

Round 1

Reviewer 1 Report

The authors presented a study examining how obesity could potentially affect pregnant uterine contractility in rats.

The obtained results “bring”, in general, a lot of important information confirming the impact of HFHSD diets, potential fetal health and pregnancy maintenance, and thus fetal development, in pregnant obese rats.

The presented experimental and analytical system seems to be, in general, sufficient to demonstrate the assumed study objectives, confirm the expected effects by researchers, and partially demonstrate the potential / applicability of the conducted experimental model for this type of scientific issue.

Below, I’ve presented some minor comments, mainly of an editorial nature, that I think, improving the overall perception and readability of the manuscript.

  1. Introduction

Line: 71 - maybe it would be better to use another expression, for example, "in obese rats", instead of "in a rat model", because after the word "model" one would have to specify what kind of rat model was used. I think that the term "obese rats" may fill the gap in the definition. This suggestion is for the authors' consideration only.

  1. Materials and Methods

2.1. Housing and handling of the animals

  1. Line: 96 - In the description of blood donation, I would consider adding a brief reference to what exactly the blood was collected, and whether the sterility conditions were maintained (I mean the fact that the risk of any chemical factor influencing the blood parameters subsequently measured is minimized).

Alternatively, the authors may consider moving this section to a separate subchapter in the Material and Methods section. This is a suggestion for reflection only.

  1. Please consider to highlight separately what happened to pregnant females from whom the uterus was collected for qPCR (as mentioned in subsection 2.4. (Line: 148), and WB analyzes (subsection 2.5.). Was it just a biopsy or was excised with a scalpel while maintaining sterility?.

2.4. RT-PCR studies

  1. Line: 140 - In the description of primers for genes - for the sake of clarity of the content reception, it would be good to enter which genes these abbreviations refer to (as was mentioned i.e. in description of figures, for example 5 or 6, etc.)
  2. Please mention in one sentence whether the results of fluorescence changes were normalized to the standard curves for the studied genes of interest, separately) or using the delta delta Cq methodology?

2.6. Uterine contractility studies

  1. Please complete the information were the following reagents came from: oxytocin, prostaglandin F2, kisspeptin (KISS1 94-121 fragment), leptin, and adiponectin.

  1. Results

3.1. Effect of HFHSD on body and organ weights, glucose tolerance and fat distribution in pregnant rats

  1. Line: 293, Fig. 3. - Here, in the description, I kindly ask you to specify whether the rats were pregnant from the 10th week or from the 9th week of age (according to the description in line 228)?
  2. Description of Figure 3. Please clearly indicate the control for which the cited statistical significance was observed for the investigated group of rats.

3.3. Effect of HFHSD on contractility of 22-day pregnant rat uterus in vitro

  1. Figure 9 (c). it is partially "cut" but, as I understand, this is the result of the text formatting at the editorial stage?.

  1. In general, in the description of all figures, wherever the levels of mRNA and proteins were quantified, according to the reviewer, for the sake of clarity of the figure description, it is worth adding information about the number of "n" individuals in individual groups of animals. Such information could be helpful in the explanation to the reader that the number of n individuals investigated for mRNA did not change from the number of individuals where changes were measured for their corresponding protein levels.
  2. The same applies to the principle that under all figures, it is worth entering, at each level of statistical significance, what was the control for the discussed parameter in a given group - for example, using the notation "[...] vs. [...] ". In other words, please consider indicating the control for which the cited statistical significance was observed for the investigated group of rats. I only submit this to the authors for consideration.
  3. In the appendix - it may be worth considering the improvement of the quality of Western-blots, the description of the individual electropherogram paths of the Western-blot analysis, eliminating, as I understand, the hand-made blots on the membrane or indicating the important positions of the molecular mass in the position of the mass marker. I also leave this issue to the decision of the journal's editors.

Author Response

Thank you for reading and reviewing our manuscript, which help us to improve its quality. We have revised our manuscript, and several changes have taken place. Please find our detailed responses below.

Line: 71 - maybe it would be better to use another expression, for example, "in obese rats", instead of "in a rat model", because after the word "model" one would have to specify what kind of rat model was used. I think that the term "obese rats" may fill the gap in the definition. This suggestion is for the authors' consideration only.

The expression has been changed to “in obese rats” in the revised manuscript.

Line: 96 - In the description of blood donation, I would consider adding a brief reference to what exactly the blood was collected, and whether the sterility conditions were maintained (I mean the fact that the risk of any chemical factor influencing the blood parameters subsequently measured is minimized).

Further information has been added to the revised manuscript about the minimizing of the risk of contamination.

Please consider to highlight separately what happened to pregnant females from whom the uterus was collected for qPCR (as mentioned in subsection 2.4. (Line: 148), and WB analyzes (subsection 2.5.). Was it just a biopsy or was excised with a scalpel while maintaining sterility?

Since the rats were sacrificed before the removal of the uterine tissues, therefore we modified the text both in PCR and Western blot sections. Each section initiated with the following: “After sacrifice and excision” making clear that the animals were not survivors.

 Line: 140 - In the description of primers for genes - for the sake of clarity of the content reception, it would be good to enter which genes these abbreviations refer to (as was mentioned i.e. in description of figures, for example 5 or 6, etc.)

The abbreviations have been clarified, while caspase 3 primer has been deleted, because it is not involved into this manuscript.

Please mention in one sentence whether the results of fluorescence changes were normalized to the standard curves for the studied genes of interest, separately) or using the delta delta Cq methodology?

The fluorescence changes were normalized using delta delta Cp method. This sentence has been added to the revised version.

Please complete the information where the following reagents came from: oxytocin, prostaglandin F2, kisspeptin (KISS1 94-121 fragment), leptin, and adiponectin.

The sources of the above-mentioned compounds have been inserted into the revised manuscript.

Line: 293, Fig. 3. - Here, in the description, I kindly ask you to specify whether the rats were pregnant from the 10th week or from the 9th week of age (according to the description in line 228)?

As it mentioned in the legend of Figure 3, the rats become pregnant at week 10. However, in line 228 there was an error, we modified 9 to 10 weeks of age, thank you to point to this mistake.

Description of Figure 3. Please clearly indicate the control for which the cited statistical significance was observed for the investigated group of rats.

The statistical significance represents the differences between the SD and HFHSD groups at the same weeks of age. This sentence has been added to the legend of Figure 3.

 Figure 9 (c). it is partially "cut" but, as I understand, this is the result of the text formatting at the editorial stage?

Yes, you are right, now there is no missing part of Figure 9.

In general, in the description of all figures, wherever the levels of mRNA and proteins were quantified, according to the reviewer, for the sake of clarity of the figure description, it is worth adding information about the number of "n" individuals in individual groups of animals. Such information could be helpful in the explanation to the reader that the number of n individuals investigated for mRNA did not change from the number of individuals where changes were measured for their corresponding protein levels.

The ”n” numbers have been added to the RT-PCR and Western blot figures.

The same applies to the principle that under all figures, it is worth entering, at each level of statistical significance, what was the control for the discussed parameter in a given group - for example, using the notation "[...] vs. [...] ". In other words, please consider indicating the control for which the cited statistical significance was observed for the investigated group of rats. I only submit this to the authors for consideration.

Thank you for the recommendation. Since we think that the Figure legends along with the curves or columns clearly state that which groups were compared statistically (except Figure 3 that has been already corrected), we do not want to apply further modifications in the Figure legend. We hope that this is acceptable for the reviewer.

In the appendix - it may be worth considering the improvement of the quality of Western-blots, the description of the individual electropherogram paths of the Western-blot analysis, eliminating, as I understand, the hand-made blots on the membrane or indicating the important positions of the molecular mass in the position of the mass marker. I also leave this issue to the decision of the journal's editors.

We have added the numerical molecular weights to the M part. We don’t want to publish the supplementary figure, but just to prove that the WB measurement was carried out correctly.

Reviewer 2 Report

In this paper authors examine the uterine contractility, cervical resistance and the expression of sex hormones, adipokines and cytokines in pregnant rats after high fat high sucrose diet (HFHSD). Methodologically well-done paper with some small issues.

  1. Justify better the choice of measured hormones, adipokines and cytokines (even they are adequately described in Discussion section).
  2. Line 179. Spell the abbreviation SPEL.
  3. Methods, Line 195-196. The white line in the methods. It looks like the part of the text is missing. The next phrase starts with “After incubation, …” May be the missing part was about this “incubation”?
  4. Fig 1, Fig 3the line numbers are shifted in the Figure space
  5. Fig 4 and Fig 9 are partially out of page readable space.
  6. Fig 8 vs text line 350. In the text 40% decrease is declared, but in the Figure 8 the AUC data columns are 180’000 vs 150’000, the difference is not 40%!
  7. Line 449, pg 13 Results: authors write that IL-1beta, IL-6, IFN gamma were not detectable. Please, comment “why”? Even background level was not detectable? If the measurement method is right?
  8. Supplemental data. If these data will be published, they need to be presented better, with explicit figure legends. Molecular weight markers (signed as M) must to have numerical molecular weight in kDa, now absent.

Author Response

Thank you for reading and reviewing our manuscript, which help us to improve its quality. We have revised our manuscript, and several changes have taken place. Please find our detailed responses below.

Justify better the choice of measured hormones, adipokines and cytokines (even they are adequately described in Discussion section).

Thank you for the comment, however it is not clear what the reviewer means on “better” justification. In the Discussion we gave a detailed description and interpretation of these factors and their impact on the obesity-induced modifications e.g., leptin is inflammatory, adiponectin is anti-inflammatory, kisspeptin is a neuropeptide that becomes adipokine during obesity and has a role in the control of reproductive processes, cytokine panel investigation reflects the inflammatory-anti-inflammatory status, and female sex hormones has crucial impact on the uterine contractility. These facts justify the choice of these parameters to be investigated and all these pieces of information are already involved in the manuscript.

Line 179. Spell the abbreviation SPEL.

SPEL is the name of the software development company that contributed to the creation of the measuring software.

Methods, Line 195-196. The white line in the methods. It looks like the part of the text is missing. The next phrase starts with “After incubation, …” May be the missing part was about this “incubation”?

There is no missing text, the white line is the initiation of the next paragraph, as everywhere in text. However, the reviewer is right because it was not clear what the incubation period was. In a bracket we put the incubation period expression to clarify this problem in the revised manuscript.

Fig 1, Fig 3the line numbers are shifted in the Figure space

This is an editorial problem that we cannot manage but thank you for the comment.

Fig 4 and Fig 9 are partially out of page readable space.

The problem has been solved in the revised version.

Fig 8 vs text line 350. In the text 40% decrease is declared, but in the Figure 8 the AUC data columns are 180’000 vs 150’000, the difference is not 40%!

The reviewer is right, the correct decrease is 16%, the value has been modified in the revised manuscript.

Line 449, pg 13 Results: authors write that IL-1beta, IL-6, IFN gamma were not detectable. Please, comment “why”? Even background level was not detectable? If the measurement method is right?

The measurement method was ultrasensitive (basically the most sensitive that is available), there was no problem with the background level, these cytokines simply were below the detection level. The measurement was done after 6 month of sample collections (a half a year was necessary to organize the measurement process in Pasteur Institut), and this delay might reduce the amount of cytokines despite careful storage and transport. Majority of the cytokines were detectable by the cytokine panel analysis suggesting that the initial concentrations of the “missing” cytokines probably were very low and they had no significant impact on the processes.

Supplemental data. If these data will be published, they need to be presented better, with explicit figure legends. Molecular weight markers (signed as M) must to have numerical molecular weight in kDa, now absent.

We have added the numerical molecular weights to the M part. We don’t want to publish the supplementary figure, but just to prove that the WB measurement was carried out correctly.